# Optimization of Polyarginine-Conjugated PEG Lipid Grafted Proliposome Formulation for Enhanced Cellular Association of a Protein Drug

**DOI:** 10.3390/pharmaceutics11060272

**Published:** 2019-06-11

**Authors:** Amolnat Tunsirikongkon, Yong-Chul Pyo, Dong-Hyun Kim, Sang-Eun Lee, Jeong-Sook Park

**Affiliations:** 1College of Pharmacy, Chungnam National University, 99 Daehak-ro, Yuseong-gu, Daejeon 34134, Korea; amolnat.tun@gmail.com (A.T.); himchani46@naver.com (Y.-C.P.); dong_bal@naver.com (D.-H.K.); nnininn@hanmail.net (S.-E.L.); 2Division of Pharmaceutical Sciences, Faculty of Pharmacy, Thammasat University, Rangsit Center, Pathumthani 12120, Thailand

**Keywords:** poly-l-arginine, DSPE-PEG, cationic proliposome, cellular association, cytotoxicity

## Abstract

The purpose of this study was to develop an oral proliposomal powder of protein using poly-l-arginine-conjugated 1,2-distearoyl-sn-glycero-3-phosphoethanolamine-poly(ethylene glycol) (DSPE-PEG) (PLD) for enhancing cellular association upon reconstitution and to compare its effects with a non-grafted and PEGylated formulation. Cationic proliposome (CATL), PLD-grafted CATL (PLD-CATL), PEGylated CATL (PEG CATL), and PLD grafted-PEG CATL (PLD-PEG CATL) were prepared and compared. Successful conjugation between poly-l-arginine and DSPE-PEG was confirmed by ^1^H NMR and FT-IR. PLD was successfully grafted onto the proliposomal powder during the slurry process. Although reconstituted liposomal sizes of CATL and PLD-CATL were increased by agglomeration, PEGylation reduced the agglomeration and increased the encapsulation. The viabilities of cells treated with both CATL and PLD-CATL formulations were low but increased following PEGylation. With regard to cellular association, PLD-CATL enhanced cellular association/uptake more rapidly than did CATL. Upon PEGylation, PEG CATL showed a lower level of cellular association/uptake compared with CATL while PLD-PEG CATL did not exhibit the rapid cellular association/uptake as seen with PLD-CATL. However, PLD-PEG CATL still enhanced the higher cellular association/uptake than PEG CATL did without PLD. In conclusion, proliposomes with PLD could accelerate cellular association/uptake but also caused high cellular toxicity. PEGylation reduced cellular toxicity and also changed the cellular association pattern of the PLD formulation.

## 1. Introduction

There has been great progress in liposomal research with the development of innovative liposomal technologies, which can be used as specific drug delivery systems. Examples of these technologies include photo-triggered [1], pH-triggered [2,3], or thermo-triggered [4] release of drugs from liposomes. Beyond its main purpose as a specific drug delivery system, liposomes have been used to deliver multiple drugs synchronously [5], reduce drug cellular toxicity [6,7], and enhance drug bioavailability [8].

Of particular note, liposomes have low stability in their aqueous dosage form. This is the main issue that limits liposomal shelf-life, which can be caused by the fusion of liposomes or leakage of encapsulated substances after long-term storage [9,10,11]. Other processes to solidify liposomes, such as freeze drying or spray drying, also introduced stress-induced instability into the liposomal system [12]. Additionally, the low cellular uptake/association of liposomes, which is observed with anionic liposomes particularly, remains a classical problem that requires a cellular association enhancer [13,14], although negative-charged systems have been approved in clinical practice using liposomal formulations such as Doxil/Caelyx. However, for oral delivery, it was reported that positively charged proliposomes could increase oral bioavailability depending on the physicochemical characteristics [15,16].

The use of a dry liposomal formulation, known as proliposomes [17], is a potential option for overcoming the stability issues of aqueous liposomes, as well as the stress-induced instability of liposomes created by other solidification methods [18]. Proliposomes could be prepared in solid dosage form using simple manufacturing methods such as granulation [19] or slurries [20] and then reconstituted into liposomes after contact with a biological fluid. However, the advance in proliposomal research has lagged in comparison with liposomal research. Most proliposomes are prepared from anionic lipids [8,21,22,23], such as phosphatidylcholine, and express a negative surface charge unless modified [24]. Thus, the main obstacle in using a proliposomal dosage form is cellular association.

The cellular association of anionic lipids depends on the type of lipid used. These anionic lipids are generally not responsive to cellular association and adhesion [13,14,25] and require some modification [26]. Therefore, we proposed that using cationic lipids to create proliposomes would presumably enhance cellular association. Cell-penetrating peptides (CPPs), in general, are short peptides consisting of amino acids that enhance the cellular adsorption of their attached cargo via various physiological pathways, such as endocytosis and non-endocytosis [27,28,29], or proteoglycans on the cell surface [30]. All pathways help increase the association of CPP-attached cargo with cells. Many types of CPPs such as transactivator of transcription (TAT) peptide [31], penetratin [32,33], end-binding protein 1 (EB1) [32], MPG (SG-GALFLGFLGAAGSTMGAWSQPKKKRKV) [32], RIPL (IPLVVPLRRRRRRRRC) [34], and polyarginine [35,36] have been found to enhance cellular association.

As mentioned, cationic lipids and CPP have certain benefits. When used to prepare a proliposomal dosage form, these benefits help overcome the classical obstacles seen with the use of liposomes. However, a synergist effect of cationic lipids and CPP on positively charged density is likely, which would lead to more cellular toxicity. However, 1,2-distearoyl-sn-glycero-3-phosphoethanolamine-poly(ethylene glycol) (DSPE-PEG), has been found to reduce the cellular toxicity of drugs and particles due, in part, to its steric effect that shields the surfaces of these particles [37,38,39,40]. Therefore, combination of cationic lipids, CPP, and PEG could increase the stability of proliposomes, reduce toxicity, and improve oral absorption.

Thus, the aim of the present study was to develop an oral cationic proliposomal powder grafted to synthesized poly-l-arginine conjugated DSPE-PEG (PLD) to enhance the cellular association of a reconstituted liposome-encapsulated protein drug, and to investigate the feasibility of enhancement in oral absorption. The poly-l-arginine conjugated DSPE-PEG was synthesized, characterized by ^1^H NMR and ATR-FTIR, and subsequently grafted onto the selected proliposomal formulation that achieved the highest encapsulation efficiency. Physicochemical properties, cellular toxicity, and cellular association/uptake of proliposomal powder grafted to PLD, DSPE-PEG, or both PLD and DSPE-PEG were characterized and compared with non-grafted and PEGylated proliposomal formulations.

## 2. Materials and Methods

### 2.1. Materials

Poly-l-arginine hydrochloride (PLR) (molecular weight = 13,300 Da), bovine serum albumin (BSA), 3-(4,5-dimethylthiazol-2-yl)-2,5-diphenyltetrazoliumbromide (MTT), and Triton X-100 were purchased from Sigma–Aldrich Co. (St. Louis, MO, USA). The 1,2-dioleoyl-3-trimethylammonium propane (DOTAP), 1,2-dioleoyl-sn-glycero-3-phosphoethanolamine-propane (DOPE), cholesterol, and 1,2-distearoyl-sn-glycero-3-phosphoethanolamine-N-[methoxy(polyethylene glycol)-2000] (DSPE-PEG) were obtained from Avanti Polar Lipids (Birmingham, AL, USA). Fetal bovine serum and Dulbecco’s Modified Eagle Medium (DMEM) were purchased from Gibco^®^ Invitrogen (Grand Island, NY, USA). 4,6-diamidino-2-phenylindole dihydrochloride (DAPI), Alexa Fluor 488, and Prolong^TM^ diamond antifade mountant were purchased from Thermo Fischer Scientific (Eugene, OR, USA). All remaining chemicals, including those used to prepare phosphate buffer saline (PBS), were of reagent grade.

### 2.2. Synthesis of Poly-l-arginine Conjugated DSPE-PEG (PLD)

The synthesis of PLD was based on a previously described method [35]. Briefly, poly-l-arginine was dissolved in 25 mL, 50 mM sodium tetraborate buffer (pH 8.5) per gram poly-l-arginine. The resulting solution was stirred vigorously for approximately 30 min and subsequently filtered through a 0.22 µm Durapore^®^ membrane (Sterile Millex GV, Sigma–Aldrich, Buchs, Switzerland) into a sterile culture tube. The appropriate stoichiometric amount (0.05 mole%) of DSPE-PEG powder was then added slowly to the solution while stirring continuously. After another 6 h of vigorous stirring at room temperature, the solution was transferred to a dialysis tube (SpectraPor^®^ dialysis tubing, molecular cutoff (MWCO)) of 6–8 kDa, Spectrum Laboratories, Inc., Rancho Dominguez, CA, USA). The synthesized product was dialyzed for 24 h in 10 mM PBS (pH 7.0), followed by an additional 24 h of dialysis in deionized water. The product was then freeze-dried for 48 h at −70 °C and a pressure of 0.2 mbar. The synthesized PLD was confirmed by ^1^H NMR in D_2_O and ATR-FTIR.

### 2.3. Preparation of Plain and Modified Cationic Proliposomal Powder by Slurry Method

Plain cationic proliposomal powder was formulated using different types of sugar and the correct amount of lipid needed to obtain the reconstituted liposomes with the highest encapsulation efficiency. Subsequently, the selected proliposomal formulation was grafted to the correct amount of the following modified materials: DSPE-PEG, PLD and PLD plus DSPE-PEG.

In this study, proliposomal powder was prepared by the slurry method, modified from Khan et al. [23]. In brief, a lipid solution was prepared from total lipids dissolved in dichloromethane as shown in Table 1, in which the composition of the various formulations was optimized from preliminary study (data not shown). Bovine serum albumin (BSA), an acidic hydrophilic protein, displays a small, concentration dependent, antioxidant effect when added to preformed liposomes [41]. As a model protein drug, BSA was loaded to encapsulate in proliposomes. Sugar powder (either sucrose or mannitol) was used as a solid carrier and mixed with BSA protein powder at the ratio shown in Table 1 until homogenous. Subsequently, this mixture was combined with the lipid solution to create a slurry. Next, the slurry was vigorously stirred under the fume hood without applying heat for 30 min to evaporate the dichloromethane. The obtained proliposomal powder was kept in a tight container and protected from light at 4 °C until use. The selected proliposomal formulation with the highest encapsulation efficiency of reconstituted liposomes was modified with lipid materials (either DSPE-PEG, PLD, or PLD plus free DSPE-PEG) that were incorporated into the lipid phase during the lipid coating process as shown in Table 2. In the case of BSA-FITC as entrapped material, the BSA was replaced with BSA-FITC to prepare the fluorescent formulation for the cellular uptake study.

### 2.4. Physicochemical Properties of Reconstituted Liposomes

#### 2.4.1. Reconstitution of Proliposomal Powder

Proliposomal powder formulations (50 mg) were reconstituted in 5 mL of two different diluents (water or PBS, pH 6.8) by shaking at 37 °C for 30 min in an incubator shaker. The obtained reconstituted liposomes were then evaluated for their physicochemical properties.

#### 2.4.2. Size and Zeta Potential

Both the particle size and zeta potential of the reconstituted liposomes were evaluated using an electrophoretic light scattering spectrophotometer (Zetasizer Nano ZS^®^, Malvern Instrument, Worcester, UK). The reconstituted liposomes were transferred to a cuvette and placed in a dynamic light scattering instrument. Data was analyzed using a software package supplied by the manufacturer. Each experiment was performed in triplicate.

#### 2.4.3. Encapsulation and Loading Efficiency

The encapsulation and loading efficiencies of the protein drug in liposomes reconstituted from proliposomal powder were determined by analyzing the filtrate after centrifugation using a 100,000 MWCO. ultracentrifuge tube (Millipore, Carrigtwohill, Ireland) at 10,000 rpm for 40 min. The protein contents in the filtrate were determined using the Bradford^®^ protein assay. The absorbance was measured at 562 nm using an ELISA plate reader (Sunrise, TECAN, Männedorf, Switzerland). The appropriate blanks for all proliposomal formulations were also prepared in order to correct the sample absorbance. Both the concentration and amount of the protein contents were calculated in comparison with a standard curve. Encapsulation efficiency was calculated by the following equation:$$\%\;\mathrm{Entrapment}\;\mathrm{Efficiency}=\frac{\left(\mathrm{Total}\;\mathrm{Protein}\;\mathrm{addition}-\mathrm{Filtrated}\;\mathrm{Protein}\right)\;\times \;100}{\mathrm{Total}\;\mathrm{Protein}\;\mathrm{addition}}$$

The loading efficiency was calculated by the following equation,
$$\%\;\mathrm{Loading}\;\mathrm{Efficiency}=\frac{\mathrm{Total}\;\mathrm{Protein}\;\mathrm{addition}\;\times \;\%\;\mathrm{Entrapment}\;\mathrm{Efficiency}}{\mathrm{Total}\;\mathrm{Lipid}\;\mathrm{addition}}$$

### 2.5. Morphology of the Proliposomal Powder and Reconstituted Liposomes by Scanning Electron Microscopy (SEM)

The morphologies of the proliposomal powder and reconstituted liposomes were observed by scanning electronic microscopy (SEM, JSM-7000F, Jeol Ltd., Tokyo, Japan). Reconstituted liposomes were air dried overnight on a cover slip before attaching to the slab and subsequently coated by the coating materials. Both proliposomal powder and reconstituted liposomes were coated with palladium using a vacuum evaporator, and examined using SEM at a 10 kV accelerating voltage.

### 2.6. In Vitro Study Using Cell Cultures

#### 2.6.1. Cell Line

Human colon carcinoma (Caco-2) cells were cultured in DMEM supplemented with 4.5 g/d,l-glucose, 10% fetal bovine serum, 1% penicillin-streptomycin, 584 mg/mL l-glutamine, and 25 mM HEPES. The culture medium was changed every 1–2 days. At all time, the cells were cultivated in an incubator at 37 °C under 5% CO_2_ and 95% relative humidity.

#### 2.6.2. Cellular Toxicity

Cell viability was assessed by 3-(4,5-dimethylthiazol-2-yl)-2,5-diphenyltetrazolium bromide (MTT) assay to determine the cytotoxicity of the formulations to the cells [42]. Caco-2 cells were first seeded at a density of 1 × 10^4^ cells/well in 96-well plates and allowed to attach overnight. The next day, the reconstituted liposomal suspensions, which were diluted in medium at concentrations ranging from 16.5 to 200 µg/mL, were added to each well at 300 µL/well to replace the old medium. After continued incubation overnight, the medium containing the samples was discarded, and 100 µL of MTT solution in medium were added to each well and further incubated for 4 h at 37 °C. Next, a dimethysulfoxide solution was added to each well, and the 96-well plate was shaken for 30 min. Finally, the absorbance was measured at 570 nm using an ELISA plate reader (Sunrise, TECAN, Männedorf, Switzerland). Untreated cells were used as a control, and their viability was set as 100%.

#### 2.6.3. Cellular Association

##### Fluorescence Activated Cell Sorting (FACS) Analysis

Cellular association was determined by FACS analysis. First, 5 × 10^5^ cells were seeded in Nunc^TM^ cell culture dishes and incubated overnight to allow for complete attachment. The next day, the medium was removed, and the reconstituted liposomal suspensions, diluted in medium to 16.5 µg/mL, were added to each well and incubated for 4 h. The cells were then harvested by trypsinization at 0, 1, 2, 3, and 4 h after incubation, respectively. Cells incubated in the absence of the liposomal suspension were used as the control. The trypsinized cells were fixed with 4% *v*/*v* formaldehyde before analyzing. The cellular association/uptake was subsequently analyzed by FACS (Beckton Dickinson, Franklin Lakes, NJ, USA). The cells were gated using forward scatter versus side scatter to exclude debris and dead cells. The represented data were expressed as mean fluorescent intensities, similar to other studies [43,44]. The data were analyzed using the FACSDiva version 6.1.3 software.

##### Confocal Laser Scanning Electron Microscope (CLSM)

The cells were visualized by CLSM (Leica, Wetzlar, Germany) to confirm cellular uptake/association. Cells were seeded on glass cover slips and placed in Nunc^TM^ cell culture dishes at the concentration of 5 × 10^5^ cells/well. The following day, cells were subsequently incubated with the liposomal suspensions at a concentration of 16.5 µg/mL for a total of 4 h. After 2 and 4 h of incubation, the cells on cover slips were washed with pure PBS, followed by PBS containing 1% Triton X-100, and then fixed with 4%v/v formaldehyde. The fixed cells were then stained for 20 min with DAPI (for nuclei) and with Alexa Fluor 488 (for actin). After staining and washing thoroughly, the glass cover slips were removed from the culture dishes and place on glass slides, which were mounted readily using anti-fade mounting gel. The cell images were visualized using CLSM within 2–3 days after sample preparation.

### 2.7. Statistical Analysis

Statistical analysis was performed using either the independent sample t-test or analysis of variance (ANOVA) in which a *p*-value < 0.05 was considered statistically significant.

## 3. Results and Discussions

### 3.1. Characterization of Synthesized Poly-l-arginine Conjugated DSPE-PEG

#### 3.1.1. ^1^H NMR

Poly-l-arginine conjugated DSPE-PEG was synthesized in-house successfully. The reaction between poly-l-arginine and DSPE-PEG was confirmed by ^1^H NMR performed at 300 MHz. The chemical shifts in deuterium oxide of the synthesized products were assigned as illustrated in Figure 1A. The spectra shown at 1.6–1.8 ppm referred to the m, –CH_2_–, and β–γ carbons of the arginine side chains, while spectra at 3.16–3.18 ppm indicated the t, ^2^H, –CH_2_–N–, and methoxy PEG linked to arginine. The spectra at 3.6 ppm and 4.3 ppm approximately, designated ^77^H, PEG and ^1^H, –N–CHR–COO–, respectively. These results corresponded to our research on poly-l-arginine and DSPE-PEG conjugation conducted and described previously [35].

#### 3.1.2. ATR-FTIR

The conjugation between poly-l-arginine and DSPE-PEG was also further confirmed by ATR-FTIR. The spectra of DSPE-PEG, poly-l-arginine, and PLD are shown in Figure 1B. The spectra of DSPE-PEG revealed the characteristic carbonyl ketone peak at around 1700 cm^−1^, as well as the C-H stretching peaks at approximately 2900 and 2800 cm^−1^ (specifically 2915.5 and 2885.1 cm^−1^), which corresponded to results described previously [45]. Regarding the spectra of poly-l-arginine, the observed peaks at around 1648 cm^−1^ and 1541 cm^−1^ indicated amide I (carbonyl C=O and guanidine C=N stretching peak) and amide II (C–N stretching peak and N–H bending peak), respectively, while the peak assigned at around 3300 cm^−1^ corresponded to the N–H stretching vibration. These results corresponded to other research findings [21,46]. For the spectrum of PLD, which was the poly-l-arginine conjugated DSPE-PEG spectra, the carbonyl ketone peak of DSPE-PEG, as well as the amide I and II peaks of poly-l-arginine, noticeably disappeared, indicating chemical interactions between DSPE-PEG and poly-l-arginine after conjugation.

### 3.2. Proliposomal Powder

#### 3.2.1. Plain-Cationic Proliposomal Powder: Effect of Lipid Amount and Type of Sugar on the Encapsulation Efficiency of Reconstituted Liposomes

The encapsulation efficiencies of liposomes reconstituted from plain-cationic proliposomal powders are shown in Figure 2. As mentioned previously, these powders were prepared using different amounts of lipids and types of sugars. With regard to the effect of sugar type on the encapsulation efficiency of reconstituted liposomes, mannitol significantly enhanced the encapsulation efficiency of reconstituted liposomes compared with sucrose (*p* < 0.05, *t*-test).

Since sucrose is a disaccharide and mannitol is a polyalcohol, these two sugars affect lipid fluidity differently. A disaccharide has the ability to insert itself into adjacent lipid head groups and form hydrogen bonds via its hydroxyl groups. Consequently, the saturated furan and pyran rings of a disaccharide structure spread apart the lipid head groups, which results in gel-phase destabilization of lipids during reconstitution. This effect depends vastly on the sugar to lipid ratio and the amount of water in the system, as described previously [47]. On the other hand, mannitol (the polyalcohol sugar) contains no ring and, as such, has less effect on lipid packing dissemination compared with sucrose. As a result, a higher encapsulation efficiency was observed with mannitol. The membrane fluidity of polyalcohols is affected by the number of hydroxyl groups. More hydroxyl groups have been shown to decrease the phase transition temperature more substantially [48].

The increase in the lipid ratio by weight from 1 to 1.5 also significantly enhanced the encapsulation efficiency of reconstituted liposomes (*p* < 0.05, *t*-test). The high encapsulation efficiency of liposomes was in direct proportion with high levels of lipid, due to the more internal volume available for materials to be encapsulated. The available internal volume was a consequence of the larger population of liposomes formed during reconstitution with regard to more lipids [49].

Using these results, we selected the appropriate proliposomal formulation for further PEGylation or modification with PLD. Specifically, we used the formulation prepared with mannitol as a carrier and coated with lipid at the ratio by weight of 1.5. This formulation is referred to as CATL here. This formulation was selected, since it obtained the highest encapsulation efficiency of all reconstituted liposomes.

#### 3.2.2. Modified-Cationic Proliposomal Powder: Physicochemical Properties of Reconstituted Modified-Liposomes

The physicochemical properties of liposomes reconstituted from cationic proliposomal powder which were grafted to DSPE-PEG, PLD, or PLD plus free DSPE-PEG, are illustrated in Table 3. PEGylation in this study was accomplished using grafted DSPE-PEG.

##### Particle Size

After reconstitution in water, the cationic liposomes were 300.6 ± 4.7 nm in size, as shown in Table 3. Upon incorporation of 0.5% DSPE-PEG and 5% DSPE-PEG into the cationic proliposomal powder, the size of the reconstituted liposomes decreased by 1.12- and 1.42-fold, respectively. These results indicated the effect of DSPE-PEG on the size reduction of reconstituted liposomes. The size of PLD-CATL reconstituted in water was slightly larger than that seen in the CATL formulation. The size of PLD-PEG CATL was reduced by 1.62-fold compared with PLD-CATL, which also confirmed the effect of DSPE-PEG on size reduction. A similar trend in size was also observed in fluorescent proliposomal formulations. After the addition of DSPE-PEG, the sizes of fluorescent CATL and PLD-CATL liposomes were decreased from 459.0 ± 14.1 nm to 451.9 ± 14.4 nm and from 465.8 ± 26.8 nm to 415.6 ± 6.71 nm, respectively. Enhancement in oral absorption of nanoparticles can potentially be achieved by altering the nanoparticle size and charge [50].

Proliposomes were also reconstituted in PBS, pH 6.8, to observe the effect of diluent-containing salts on liposomal agglomeration. Reconstituted CATL and PLD-CATL were remarkably larger compared with those reconstituted in water. After the addition of 0.5% and 5% DSPE-PEG to CATL, the size was reduced by 1.38-fold and 10.08-fold, respectively. After the addition of 5% DSPE-PEG to PLD-CATL, the size was decreased by 7.61-fold, which indicated a very strong protective effect of DSPE-PEG on the agglomeration of liposomes in the diluent-containing salt. This decrease in size occurred because of the effect of steric hindrance of DSPE-PEG [51], which caused lower agglomeration even in the diluent-containing salts.

Due to the important effect on both the physical and chemical stability of proteins, pH is one of the critical factors in formulation of proteins as well as stability in gastric pH of protein drugs is very important. Generally, buffer systems could mediate protein stabilization depending on proper pH ranges for each protein [52]. However, it is suggested that proliposomes are stable in various pH conditions of gastrointestinal tract. It was reported that some drugs that could not be well absorbed could be entrapped in liposomes to increase the rate and the amount of the absorption in gastrointestinal tract. Furthermore, it can also increase the stability and absorption rate of the peptide drugs [15,53]. In addition, our previous study showed that proliposomes enhanced oral bioavailability and stability of antioxidant peptide, glutathione [42]. Moreover, protective polymers such as PEG could impart stability and long-circulating properties [54]. Therefore, these proliposome formulations could be stable in various pH conditions of gastrointestinal tract and increase the oral absorption.

The size and morphology of reconstituted liposomes in diluent-containing salts were also confirmed by SEM images, as described in a further section.

##### Zeta Potential

Upon reconstitution, the charge of CATL was 44.80 ± 1.60 mV. This positive surface charge was a result of the cationic lipids coated onto the surface of proliposomal powder. The zeta potential decreased to 21.90 ± 1.16 mV after addition of 0.5% DSPE-PEG and to −0.23 ± 0.15 mV after addition of 5% DSPE-PEG, indicating the influence of DSPE-PEG on the reduction of zeta potential and also on the reduction of size, as mentioned in the above. In PLD-CATL, the zeta potential of the reconstituted liposomes was positive, 44.17 ± 2.87 mV, but decreased after the addition of DSPE-PEG (PLD-PEG CATL formulation) to −1.05 ± 1.04 mV. The similar tendency of zeta potential was also observed in fluorescent proliposomal formulations. The surface charge of fluorescent CATL liposomes decreased from 32.6 ± 1.30 mV to −0.62 ± 0.18 mV after the addition of 5% DSPE-PEG and from 31.73 ± 2.89 mV to 0.26 ± 0.16 mV for PLD-CATL and 5% DSPE-PEG addition into PLD-CATL, respectively.

DSPE-PEG, even at the relatively low grafting densities, exhibited a near-neutral zeta potential of particles regarding its shielding effect [39,40]. Since DSPE-PEG is a hydrophilic-uncharged polymer, the particles surface expressed the close to neutral charge by the shielding effect of DSPE-PEG. PLD is the conjugated product of poly-l-arginine and DSPE-PEG. It expresses a positive charge related to the structure of poly-l-arginine. After the addition of PLD into cationic lipids, the charge remained positive. However, the charge decreased in the presence of free DSPE-PEG on the particle surface, also indicating the effect of DSPE-PEG on the surface charge of particles.

##### Encapsulation and Loading Efficiency

The encapsulation efficiencies of all formulations were noticeably increased following the addition of DSPE-PEG (in proportion to the amount of DSPE-PEG added). The PLD-CATL and PLD-PEG CATL formulations obtained higher encapsulation efficiencies of reconstituted liposomes compared with PEG CATL. Encapsulation efficiency is expressed in Table 3 as the efficiency percentage (%EE).

DSPE-PEG grafted onto the lipid surface substantially reduced the permeation of encapsulated materials from the inside. When grafting DSPE-PEG onto the lipid surface, particles were shielded by the steric effect of DSPE-PEG, and hydration of the outer layer was increased in association with dehydration of the lipid head group region by DSPE-PEG [55,56]. As a result, the stability of the lipid layer increased. From this hypothesis, retention of the protein drug could be higher using a more stabilized lipid in conjunction with DSPE-PEG. Thus, in this study we observed that all formulations including DSPE-PEG expressed high encapsulation efficiencies.

##### Morphology of Proliposomal Powder and Reconstituted Liposomes by SEM

The morphologies of proliposomal powder and reconstituted liposomes were observed under SEM, as shown in Figure 3. To determine and confirm the actual size of reconstituted liposomes, proliposomal powder was reconstituted in PBS, pH 6.8. Surprisingly, the reconstituted liposomes in all formulations ranged in size from 200 to 300 nm, similar to those reconstituted in water (Table 3), and with a similar degree of particle agglomeration. CATL was agglomerated and became less agglomerated in the formulation after addition of 5% DSPE-PEG. A similar effect was seen with the PLD-CATL formulation in that PLD-CATL liposomes were agglomerated and then became less agglomerated after incorporation of 5% DSPE-PEG.

SEM also revealed the morphology of proliposomal powder. In all formulations, mannitol powder, displaying a rod shape, was clustered together by the coated lipid. This observation was relatively similar for all formulations.

Therefore, DSPE-PEG effectively reduced the agglomeration (i.e., size) of reconstituted liposomes in the diluent that contained some salts. This was advantageous in controlling the agglomeration of liposomes under physiological conditions. This benefit was observed in both diluent-containing salts and water, although to a slightly lesser extent in the latter. Size was measured using the Zeta-sizer^®^. The main benefit of DSPE-PEG was to shield the surface of liposomes from aggregation, which was a result of its steric hindrance effect [40,55,57].

### 3.3. In Vitro Study in Cell Culture

#### 3.3.1. Cellular Toxicity

The cytotoxicity of liposomes reconstituted from proliposomal powder was evaluated in Caco-2 cells as shown in Figure 4. The doses evaluated were in the range of 16.5–200 µg/mL. The pure protein drug was used as a reference. The loss of cell viability was in relation to the concentration as well as the composition of the formulations at all applied doses.

The addition of the CATL formulations to cells resulted in the considerable loss of cell viability, especially with increased doses. However, cell viability was improved by the PEG CATL formulations, in conjunction with increasing amount of DSPE-PEG. Cell viability reached 90%−100% in the formulation after 5% DSPE-PEG addition. However, the dose could not exceed 100 µg/mL even in the formulation containing 5% DSPE-PEG.

The cell viability was relatively low after incubation with PLD-CATL compared with CATL, since both PLD and CATL express a positive charge. Similar to the CATL formulation, the cell viability after PLD-CATL incubation was also enhanced by the addition of free DSPE-PEG into the formulation. The cell viability of the PLD-PEG CATL formulation was more pronounced than PDL-CATL formulation. However, the dose could not exceed 50 µg/mL even in the formulation containing 5% DSPE-PEG. At a low concentration of the protein drug (16.5 µg/mL), both CATL and PLD-CATL formulations showed a similar, acceptable cell viability at approximately 65%.

Cell viability was evidently related to the zeta potential of the reconstituted liposomes in this study. From the results shown in Table 3, the less dominant positively charged liposomes resulted in a higher percentage of viable cells. This was observed in the liposomal formulations incorporating DSPE-PEG. Lower cell viability was observed in the CATL and PLD-CATL formulations, which expressed a positive charge.

The high charge density of the cationic lipid species resulted in the more electrostatic interactions with cells. This resulted in the cellular toxicity, that was clearly observed in in vivo studies [58,59]. However, similar to some study [38], we demonstrated a protective effect of DSPE-PEG. In our study, DSPE-PEG was grafted onto the cationic lipid species of proliposome to preserve cell viability after proliposomal reconstitution and incubation with cells.

#### 3.3.2. Cellular Association/Uptake According to FACS and CLSM

##### Fluorescence Activated Cell Sorting (FACS)

The cellular association behaviors of reconstituted liposomes were evaluated in Caco-2 cells, since these formulations are intended to be taken orally. Reconstituted liposomal formulations were incubated with cells for a total of 4 h and the cellular association at 1, 2, 3, and 4 h were investigated by FACS (Figure 5). PLD grafted onto CATL (PLD-CATL) decreased the time needed to achieve the maximum cellular association/uptake of CATL to 2 h, at 2.4-fold higher than that compared with CATL alone. However, the uptake time associated with PLD-CATL was dramatically decreased from 3–4 h compared with the formulation without PLD (CATL). This was likely due to permeation of the formulation after time.

After incorporation of DSPE-PEG into the CATL formulation (PEG CATL), the cellular association/uptake was slightly decreased at all time points evaluated. However, incorporation of PLD into this PEG CATL formulation (PLD-PEG CATL) could enhance the initial cellular uptake and continuously enhanced the cellular uptake. This was observed at all time points compared with the PEG CATL formulation.

CPP contains a number of amino acid residues that could efficiently deliver the associated cargo into cells. Poly-l-arginine is a CPP that shares a similar cellular uptake (internalization) pathway with those of other CPPs [60,61]. As such, poly-l-arginine can also deliver the associated cargo or particles into cells. Various cellular internalization pathways of CPP have been evaluated and proposed by several research groups. These include endocytosis or non-endocytosis pathways [28] or those involving cell surface glycosaminoglycans [62]. However, it is possible that more than a single process is involved in cellular association (e.g., the electrostatic interactions between the lipid membrane and cationic CPP). Together, these processes could boost the cellular uptake/association of CPP-associated molecules/particles.

In this study, the PLD-CATL formulation expressed CPP (poly-l-arginine) on its surface and had the rapid effect of enhancing Caco-2 cellular uptake. This formulation did not include the free DSPE-PEG on its surface and, therefore, it could instantaneously undergo cellular internalization via CPP. Thus, cellular uptake was more rapid with PLD-CATL than PEG CATL or CATL formulations. However, the cellular uptake pathway should be evaluated further.

For PLD-PEG CATL, the shielding effect caused by the long-steric chain of free DSPE-PEG could possibly reduce the activity of PLD on the liposomal surface. This result was confirmed by the reduction in the positively charged density of PLD-CATL. Thus, the cellular uptake efficiency of PLD-PEG CATL was lower than that of PLD-CATL. In addition, both formulations expressed a different uptake pattern. The rapid uptake pattern was not anymore observed in PLD-PEG CATL as in PLD-CATL.

PEGylation associated with a lower level of cellular association/uptake due to the inhibition of CATL or PLD-CATL cell surface attachment, which was caused by the steric chain of DSPE-PEG. In fact, DSPE-PEG did not inhibit cellular attachment via electrostatic interactions. However, DSPE-PEG severely reduced the effective surface potential [63] and, as a result, decreased the cellular association of liposomes.

The conformation of DSPE-PEG after grafting onto the particles changed with respect to concentration. These conformations influenced the cellular uptake efficiency. Yang and colleagues evaluated both the conformation and density of PEG after grafting with respect to cellular uptake and found that the stealth property was reached at the PEG grafting density that achieved brush conformation [39]. When >4 mole% DSPE-PEG2000 was added to the lipids, either the blush conformation or the transition to blush conformation was observed [40,64,65]. Thus, with the addition of 5 mole% addition in our study, the conformation was predicted to be either blush mode or the transition to blush mode. These were the points at which the stealth effect was achieved, which possibly inhibited cellular association/uptake.

##### CLSM

These cellular association behaviors were also confirmed by CLSM assay. The confocal microscopy study on Caco-2 cells incubated with liposomes for 2 h and 4 h was performed as shown in Figure 6A,B, respectively. The blue and red colors represent nucleus and actin filament staining, respectively. The same tendency of cellular association/uptake was observed by CLSM as that in the FACS results. The green color represented the fluorescent particles that could be taken by cells, which was more pronounced after 4 h versus 2 h of incubation for all formulations except PLD-CATL. For PLD-CATL, the association/uptake of fluorescent particles was noticeably higher at 2 h but decreased at 4 h of incubation. This study confirmed the benefit of PLD on enhancing the cellular association of reconstituted liposomes, revealing a different pattern when free DSPE-PEG was present.

## 4. Conclusions

Poly-l-arginine-conjugated DSPE-PEG was grafted onto cationic proliposomal powder, and its activity after reconstitution was evaluated. The conjugation of poly-l-arginine to DSPE-PEG occurred between the carbonyl ketone of DSPE-PEG and the amide groups of poly-l-arginine. Plain cationic proliposomes prepared with DOTAP, DOPE, and cholesterol as the coated lipid were more suitable with mannitol as the solid carrier rather than sucrose. Mannitol was used because of its higher encapsulation efficiency in this system compared with sucrose. The addition of free DSPE-PEG into cationic proliposomes markedly decreased agglomeration of the reconstituted liposomes, which was observed in both PBS (pH 6.8) or water, and increased the encapsulation efficiency (best result at 5% w/w). These findings are important for controlling particle agglomeration for oral proliposomes at physiological pH.

Both CATL and PLD-CATL formulations expressed a positive charge after reconstitution and demonstrated toxicity to cells. The addition of free DSPE-PEG (PEGylation) into these two cationic formulations augmented cell viability. This knowledge can help maintain a higher level of cell viability for cationic proliposomal formulations. Cellular uptake/association occurred more rapidly for PLD-CATL compared with the CATL formulations that gradually increased over time. PEG CATL showed the lower levels of cellular uptake compared with CATL, but increased after addition of PLD (PLD-PEG CATL) at all point of time observed. Thus, the formulation that enhanced cellular uptake best while preserving cell viability was PLD with added free DSPE-PEG (PLD-PEG CATL). This formulation also exhibited a satisfactory encapsulation efficiency. This finding illustrated the combined effect of CPP and free PEG on enhancing both cell viability and cellular uptake. However, the ratio of free DSPE-PEG to PLD should be adjusted in further studies to optimize both cellular uptake and cell viability efficiencies. This invented dosage was in dried powder form, which is readily formulated for oral dosage forms, such as tablet, capsule, or enteric-coated tablet/capsule.

## Figures and Tables

**Figure 1 pharmaceutics-11-00272-f001:**
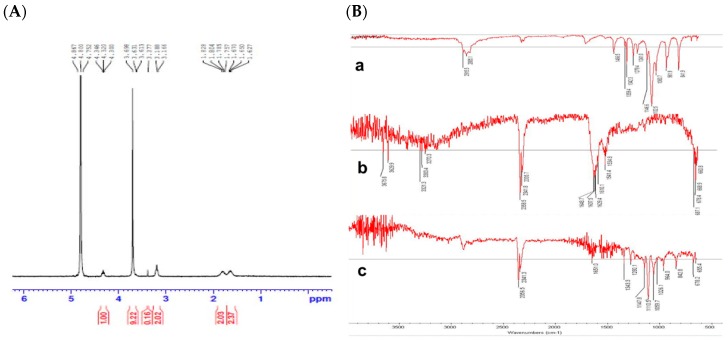
(**A**) ^1^H NMR spectra of the synthesized poly-l-arginine conjugated DSPE-PEG, (**B**) ATR-FTIR spectra of DSPE-PEG2000 (a), poly-l-arginine (b) and poly-l-arginine conjugated DSPE-PEG or PLD (c), respectively.

**Figure 2 pharmaceutics-11-00272-f002:**
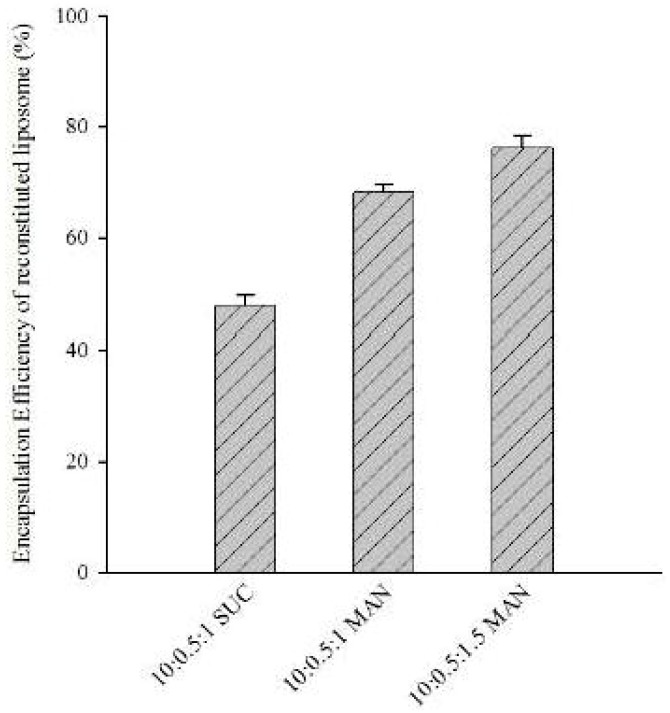
Encapsulation efficiency of liposome reconstituted from proliposomal powder prepared with different parameters (n = 3).

**Figure 3 pharmaceutics-11-00272-f003:**
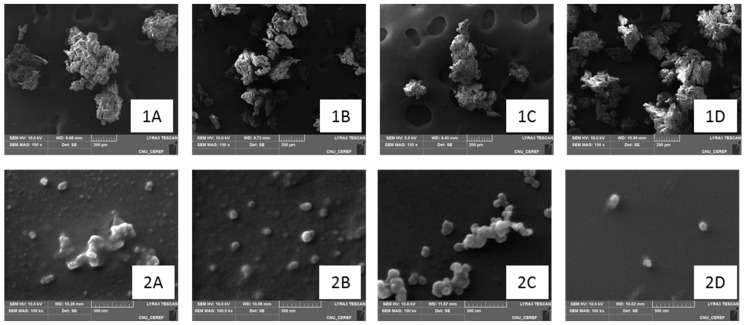
Morphology of cationic proliposome/liposome (1A/2A), PEGylated cationic proliposome/liposome (1B/2B), PLD incorporated cationic proliposome/liposome (1C/2C), PLD incorporated PEGylated cationic proliposome/liposome (1D/2D).

**Figure 4 pharmaceutics-11-00272-f004:**
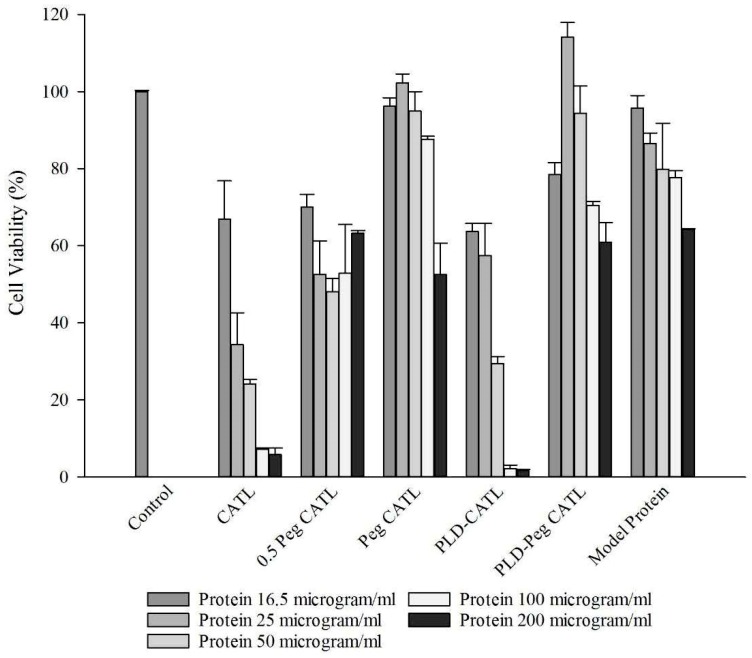
Cell viability evaluated in Caco-2 cells by MTT assay of liposome reconstituted from proliposomal powders incubated with cells at different doses ranging from 16.5 µg/mL to 200 µg/mL (n = 3).

**Figure 5 pharmaceutics-11-00272-f005:**
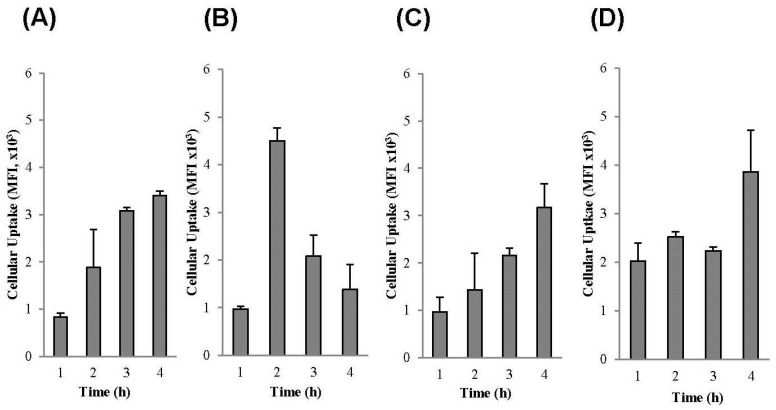
Cellular uptake evaluated in Caco-2 cells by fluorescence activated cell sorting (FACS) of liposome reconstituted from proliposomal powders incubated with cells from 1 h to 4 h (n = 3, mean ± SD). (**A**) cationic proliposome (CATL), (**B**) PEG CATL, (**C**) PLD-CATL, and (**D**) PLD-PEG CATL. In all analyses, *p* < 0.05 (**) and *p* < 0.1 (*) compared to cellular uptake at 1 h, and *p* < 0.1 (^†^) compared to maximum cellular uptake of each group.

**Figure 6 pharmaceutics-11-00272-f006:**
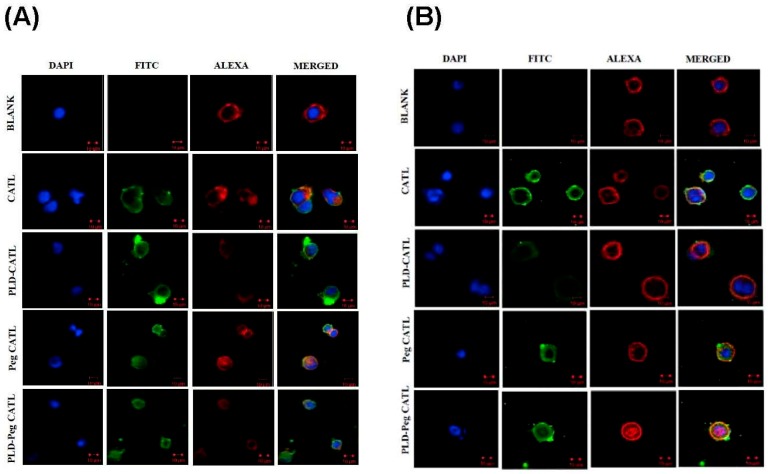
Represented images by confocal laser scanning electron microscopy (CLSM) of Caco-2 cells incubated with liposome reconstituted from proliposomal powders. Images were taken after (**A**) 2 h incubation and (**B**) 4 h incubation, respectively. Each bar represents 10 μm.

**Table 1 pharmaceutics-11-00272-t001:** Various formulations of plain cationic proliposomal powder.

Factor	Formulation (Sugar:Protein:Lipid)	Mannitol (gram)	Sucrose (gram)	Protein (gram)	Total Lipid *^a^* (gram)
Type of sugar	10:0.5:1SUC	-	10	0.5	1
	10:0.5:1MAN	10	-	0.5	1
Amount of lipid	10:0.5:1MAN	10	-	0.5	1
	10:0.5:1.5MAN *^b^*	10	-	0.5	1.5

*^a^* All formulations were prepared at fixed mole% of total lipid (DOTAP:DOPE:CHOL 75:20:5). *^b^* Formulation of 10:0.5:1.5MAN was represented as CATL in the whole experiment. Abbreviations: MAN, mannitol; SUC, sucrose.

**Table 2 pharmaceutics-11-00272-t002:** Proliposomal powder prepared with various types of modified substances.

Modified Substances (MS)	Formulation Code	Compositions (Mole%) DOTAP:DOPE:Chol:(MS)
None	CATL	75:20:5: (0)
PLD	PLD-CATL	70:20:5: (5)
DSPE-PEG	PEG CATL	70:20:5: (5)
-	0.5 PEG CATL	74.5:20:5: (0.5)
PLD plus DSPE-PEG	PLD-PEG CATL	65:20:5: (5 plus 5)

**Table 3 pharmaceutics-11-00272-t003:** Effect of modified substances; DSPE-PEG, PLD and PLD plus DSPE-PEG, grafted onto cationic proliposome (CATL) on size, zeta potential, and encapsulation efficiency of reconstituted liposome (n = 3).

Formulations	Mean ± SD						
	Size (nm)		PdI		Zeta Potential (mV)	Encapsulation (%)	Loading (%)
	DI Water	PBS, pH 6.8	DI Water	PBS, pH 6.8			
CATL	300.6 ± 4.7	3633.0 ± 286.3	0.40 ± 0.05	0.31 ± 0.12	44.8 ± 1.6	76.2 ± 2.3	25.4 ± 0.8
0.5% PEG CATL	268.3 ± 13.5	2639.7 ± 341.1	0.34 ± 0.13	0.34 ± 0.13	21.9 ± 1.2	88.3 ± 0.4 *	29.4 ± 0.1 *
PEG CATL	212.3 ± 3.7	360.3 ± 10.4	0.39 ± 0.08	0.30 ± 0.06	−0.2 ± 0.2	94.0 ± 0.6 *	31.0 ± 0.2 *
PLD-CATL	334.9 ± 7.2	2650.0 ± 183.9	0.46 ± 0.02	0.55 ± 0.12	44.2 ± 2.9	93.3 ± 0.3 *	31.1 ± 0.1 *
PLD-PEG CATL	206.7 ± 4.7	348.3 ± 1.9	0.39 ± 0.03	0.45 ± 0.01	−1.1 ± 1.0	93.8 ± 0.6 *	31.3 ± 0.2 *

* means *p* < 0.05 compared to CATL.

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
