# Peer review of "Optimization of Polyarginine-Conjugated PEG Lipid Grafted Proliposome Formulation for Enhanced Cellular Association of a Protein Drug"

_pharmaceutics, 2019, doi:10.3390/pharmaceutics11060272_

Round 1
Reviewer 1 Report
In this manuscript, authors designed a cationic preliposomal powder to improve the instability issue of liposomal suspension, and introduced poly-L-arginine-conjugated DSPE-PEG to increase the cellular uptake. The preliposomal powder was well characterized and compared among different formulations. The good reconstitution of preliposomal powder was also verified, however, the cellular studies were not well performed and presented with promising results. The detailed comments are listed as below.
The appurtenances of preliposome powder and reconstituted liposome should be provided.
Did author sterilize those preliposome powder or reconstituted liposome in cell study?
Many of the results in this manuscript are presented without statistical analysis (like table3, figure 4 and 5).
What are A, B, C and D in the figure 5?
The results in the figure 6 were poorly presented and analyzed. There are many uncertain issues in those figures. The nucleus and actin filament staining are not consistent among groups. FITC labeling of CATL is not differentiated among groups. In addition, the scale bar in figure 6 is not readable. Authors should revise carefully.
Author Response
In this manuscript, authors designed a cationic preliposomal powder to improve the instability issue of liposomal suspension, and introduced poly-L-arginine-conjugated DSPE-PEG to increase the cellular uptake. The preliposomal powder was well characterized and compared among different formulations. The good reconstitution of preliposomal powder was also verified, however, the cellular studies were not well performed and presented with promising results. The detailed comments are listed as below.
The appurtenances of preliposome powder and reconstituted liposome should be provided.
Composition of proliposomes was described in Table 2. Amount of PLD, DSPE-PEG, and PLD plus DSPE-PEG was shown as mole percentage in parenthesis, respectively.
Did author sterilize those preliposome powder or reconstituted liposome in cell study?
Thank you for kind comments. In general, sterilization process of formulation is not included in cell study. Thus, we did not sterilize proliposome powder.
Many of the results in this manuscript are presented without statistical analysis (like table3, figure 4 and 5).
Statistical analyses were performed on data in Table 3 and Figure 5. In case of Figure 4, line was added to be comparable as low toxic viability 80%~.
What are A, B, C and D in the figure 5?
Thank you for kind comments. Legend of figure 5 was revised as (A) CATL, (B) PEG CATL, (C) PLD-CATL, and (D) PLD-PEG CATL. In addition, the order of graphs was revised to make understood.
The results in the figure 6 were poorly presented and analyzed. There are many uncertain issues in those figures. The nucleus and actin filament staining are not consistent among groups. FITC labeling of CATL is not differentiated among groups. In addition, the scale bar in figure 6 is not readable. Authors should revise carefully.
Thank you for kind comments. Figure 6A and 6B were corrected and revised figure files were attached.
Reviewer 2 Report
The manuscript describes the development and characterization of proliposomes containing proteins.
The authors described a formulation developed by using cationic phospholipids, cholesterol, sugars and a derivate made up of a poly-L-arginine and DSPE-PEG for oral administration. It is unclear the rationale of using this kind of system; in fact, the developed proliposomes should require a gastro-protected oral solid formulation to be administered.
Otherwise, the authors should perform specific experiments concerning the effect of a strong acidity (like that of the stomach) on the structure and physico-chemical properties of the proposed systems.
Introduction – page 1, line 43 – “Additionally, the low cellular uptake/association of liposomes….particularly, remains a classical problem that requires a cellular association enhancer [13,14]”. The most important liposomal formulations used in clinical practice (for example Doxil/Caelyx) are negative-charged systems. The concept should be properly revised.
Page 3, line 98 – “The appropriate stoichiometric amount of DSPE-PEG powder was then added slowly to the solution while stirring continuously”. The amount of DSPE-PEG should be added.
Page 3 – section 2.3 and Table 1 – the composition of the various formulations should be justified or a suitable reference added. The choice of albumin as protein should be discussed.
Author Response
The manuscript describes the development and characterization of proliposomes containing proteins.
The authors described a formulation developed by using cationic phospholipids, cholesterol, sugars and a derivate made up of a poly-L-arginine and DSPE-PEG for oral administration. It is unclear the rationale of using this kind of system; in fact, the developed proliposomes should require a gastro-protected oral solid formulation to be administered.
Thank you for comments, the rationale of the manuscript was revised to be clear in introduction. Stability in gastric pH of protein drugs is very important. It was reported that some drugs that could not be well absorbed could be entrapped in liposomes to increase the rate and the amount of the absorption in gastrointestinal tract. Furthermore, it can also increase the stability and absorption rate of the peptide drugs (Arien et al., 1993; Xing, 2003). Moreover, our previous study showed that proliposomes enhanced oral bioavailability and stability of antioxidant peptide, glutathione (Byeon et al., 2019). Therefore, proliposomes are stable in various pH conditions of gastrointestinal tract and could increase the oral absorption.
References:
A. Arien, C. Goigoux, C. Baquey, B. Dupuy, Study of in vitro and in vivo stability of liposomes loaded with calcitonin or indium in the gastrointestinal tract, Life Sci. 53 (1993) 1279–1290.
L. Xing, Oral colon-specific drug delivery for bee venom peptide: development of a coated calcium alginate gel beads-entrapped liposome, J. C. R. 93 (2003) 293–300.
2. Otherwise, the authors should perform specific experiments concerning the effect of a strong acidity (like that of the stomach) on the structure and physico-chemical properties of the proposed systems.
Thank you for comments. Similar to the response to the first comment, stability in gastric pH of protein drugs is very important. It was reported that some drugs that could not be well absorbed could be entrapped in liposomes to increase the rate and the amount of the absorption in gastrointestinal tract. Furthermore, it can also increase the stability and absorption rate of the peptide drugs (Arien et al., 1993; Xing, 2003). Moreover, our previous study showed that proliposomes enhanced oral bioavailability and stability of antioxidant peptide, glutathione (Byeon et al., 2019). Therefore, proliposomes are stable in various pH conditions of gastrointestinal tract and could increase the oral absorption.
3. Introduction – page 1, line 43 – “Additionally, the low cellular uptake/association of liposomes….particularly, remains a classical problem that requires a cellular association enhancer [13,14]”. The most important liposomal formulations used in clinical practice (for example Doxil/Caelyx) are negative-charged systems. The concept should be properly revised.
Thank you for comments. As the reviewer pointed out, negative-charged systems are approved in clinical practice using liposomal formulations. However, for oral delivery, positively charged proliposomes increased oral bioavailability depending on the physicochemical characteristics (Arregui et al., 2018; Daeihamed et al., 2017). Therefore, the sentence was revised in focused on the oral delivery.
J.R. Arregui, S.P. Kovvasu, G.V. Betageri, Daptomycin proliposomes for oral delivery: formulation, characterization, and in vivo pharmacokinetics, AAPS PharmSciTech 19 (2018) 1802-1809. doi:10.1208/s12249-018-0989-0.
M. Daeihamed, A. Haeri, S.N. Ostad, M.F. Akhlaghi, S. Dadashzadeh, Doxorubicin-loaded liposomes: enhancing the oral bioavailability by modulation of physicochemical characteristics, Nanomedicine 12 (2017) 1187-1202. doi: 10.2217/nnm-2017-0007.
4. Page 3, line 98 – “The appropriate stoichiometric amount of DSPE-PEG powder was then added slowly to the solution while stirring continuously”. The amount of DSPE-PEG should be added.
Thank you for suggestion. Exact amount of DSPE-PEG was added as 5 mole% in revised manuscript.
5. Page 3 – section 2.3 and Table 1 – the composition of the various formulations should be justified or a suitable reference added. The choice of albumin as protein should be discussed.
Thank you for suggestion. We optimized the composition of the various proliposomal formulations from preliminary study. Therefore, the sentence was revised as follows.
Page 3, line 116-118: In brief, a lipid solution was prepared from total lipids dissolved in dichloromethane as shown in Table 1, in which the composition of the various formulations was optimized from preliminary study (data not shown).
In addition, the choice of albumin as protein was discussed in revised manuscript (page 3, line 118-120).
Bovine serum albumin (BSA), an acidic hydrophilic protein, displays a small, concentration dependent, antioxidant effect when added to preformed liposomes [Riedl et al. 1996]. As a model protein drug to encapsulate in proliposomes, BSA was loaded.
A. Riedl, Z. Shamsi, M. Anderton, P. Goldfarb, A. Wiseman, Differing features of proteins in membranes may result in antioxidant or prooxidant action: opposite effects on lipid peroxidation of alcohol dehydrogenase and albumin in liposomal systems, Redox Rep. 2 (1996) 35-40. doi: 10.1080/13510002.1996.11747024.
Round 2
Reviewer 1 Report
Fine with those responses.
No further correction is needed.
Reviewer 2 Report
The manuscript was duly revised and it can be accepted.